# Film Adaptation as Experimental Game Design

**Pippin Barr** 

Department of Design and Computation Arts, Concordia University, Montréal, QC H3G 1M8, Canada;
pippin.barr@concordia.ca

**Abstract:** Film adaptation is a popular approach to game design, but it prioritizes blockbuster films and conventional "game-like" qualities of those films, such as shooting, racing, or spatial exploration. This leads to adaptations that tend to use the aesthetics and narratives of films, but which miss out on potential design explorations of more complex cinematic qualities. In this article, I propose an experimental game design method that prioritizes an unconventional selection of films alongside strict game design constraints to explore tensions and affinities between cinema and videogames. By applying this design method and documenting the process and results, I am able both to present an experimental set of videogame film adaptations, along with potentially generative design and development themes. In the end, the project serves as an illustration of the nature of adaptation itself: a series of pointed compromises between the source and the new work.

**Keywords:** film adaptation; experimental game design; game design process documentation

## 1. Introduction

Film adaptation has had a significant history within videogame design, from early titles such as *E.T.*[1] and *Tron*[2] to contemporary games like *Alien: Isolation*[3] or *LEGO Star Wars: The Skywalker Saga*.[4] Although large-scale videogame adaptations released alongside major films have now generally been replaced by mobile-oriented casual games with contemporary films' aesthetics (e.g., *Toy Story Drop!*'s[5] content update with the release of *Toy Story 4*[6]), films and videogames have certainly walked hand in hand ([King 2002](#); [Fassone 2018](#)).

A key driving force in this pairing of media is the marketing power and brand-recognition of large-scale film production, so videogame adaptation of blockbuster films has been and continues to be the focus. While this does not always pay off (the adaptation of *E.T.*, which was literally buried in a landfill, is perhaps the most famous failed videogame of all time), it aligns well with the generally risk-averse videogame industry's desire for more or less guaranteed exposure and audience.

Blockbusters and mainstream videogame design are well-suited to one another, and adaptations further tend to select films or aspects of films that fit specifically within established videogame genres and mechanics, leading to a prevalence of action movies specifically. We have seen a predominance of combat-oriented games (e.g., *Mad Max*,[7] *LEGO's Marvel Avengers*,[8] *Friday the 13th: The Game*[9]) that

---

1   *E.T.* Directed by Stephen Spielberg. Universal Studios, 1982.
2   *Tron.* Directed by Steven Lisberger. Walt Disney Productions, 1982.
3   *Alien: Isolation*. Developed by Creative Assembly. Sega, 2014.
4   *LEGO Star Wars: The Skywalker Saga*. Developer by Traveller's Tales. Warner Bros. Interactive Entertainment, 2020.
5   *Toy Story Drop!* Developed by Big Fish Games. Big Fish Games, 2019.
6   *Toy Story 4*. Directed by Josh Cooley. Walt Disney Pictures, 2019.
7   *Mad Max*. Developed by Avalanche Studios. Warner Bros. Interactive Entertainment, 2015.
8   *LEGO's Marvel Avengers*. Developed by Traveller's Tales. Warner Bros. Interactive Entertainment, 2016.
9   *Friday the 13th: The Game*. Developed by IllFonic and Black Tower Studios. Gun Media, 2017.

adapt the big screen violence of their source material, or racing and flying games that draw on movies about motion (e.g., *Cars*[10] or *Top Gun: Hard Lock*[11]). In sum, most film adaptation into videogame form focuses on films that fit within established videogame tropes. This process has become ever more intertwined as Hollywood looks more and more toward videogames to provide film franchises (e.g., *Assassin's Creed*,[12] *Tomb Raider*,[13] *Sonic the Hedgehog*[14]) and inspiration (e.g., *Wreck-It Ralph*,[15] *Ready Player One*[16]).

Cinema is a far more diverse art form, however, than is represented by major Hollywood action blockbusters, and indeed, even blockbusters are more diverse that their simplified representations in games. In cleaving to such a limited subset of films, videogame adaptations have missed out on the significant potential of adapting elements of other genres and sources of cinema. The cinematic language, narrative structures, highly specific visual and audio moments, and countless other innovations in the larger world of film, including perennial critical favorites (e.g., *Citizen Kane*,[17] *Vertigo*[18]), films from outside the Western canon (e.g., *Touki Bouki*,[19] *An Elephant Sitting Still*[20]), and short films (e.g., *La Jetée*,[21] *Kitchen Sink*[22]), are a potential treasure trove for game design. Most of all, film would seem to represent a major opportunity for videogame designers and developers to challenge and expand on the expressive capacity of their medium. Beyond the creation of videogame adaptations specifically, cinema could be a source of innovation and rethinking in game design more broadly. In short, an enormous resource is being left largely unused.

In this article I seek to explore the question: *what potential game-design insights can be gained from adapting unconventional cinematic choices?* That is, how might the adaptation of specific examples of cinema help designers to reflect on and discover new design forms more generally? This is a question that should be answered through practice, and so my exploration will focus more or less entirely on discussion of the conceptualization, design, and development of a specific series of videogame adaptations based on diverse films from multiple genres. Although there are long-standing traditions of theory and analysis in adaptation studies (Hutcheon 2013), transmedia storytelling (Tosca and Klastrup 2016), and indeed, film-to-videogame adaptation (Fassone 2018), in this article *making* is my compass. This creation-first method is with reference to approaches such as research through design (Godin and Zahedi 2014), reflective practice (Schön 1983), and research creation (Chapman and Sawchuk 2012).

In addition to a general investigation of adaptation, I came into this project with a particular interest in contemplating videogame violence through the lens of cinema. Violence in games is and has been a hotly debated subject, with significant critical (Keogh 2012) and scientific (Ferguson et al. 2020) discussion. The base-line agonism of many game designs (and indeed many game definitions) has always lent itself to aggressive or violent forms and representations of play in ways that have come to seem ordinary and are easily accepted by players. As a game designer I am personally concerned by such trivializing treatment of violence in play, and my own design work has thus frequently touched on critiques of violence in games (e.g., *Jostle Bastard*,[23] *A Series of Gunshots*[24]). In turning to film adaptation

---

[10] *Cars*. Developed by Rainbow Studios. THQ, 2006.
[11] *Top Gun: Hard Lock*. Developed by 505 Games. Paramount Digital Entertainment, 2012.
[12] *Assassin's Creed*. Directed by Justin Kurzel. 20th Century Fox, 2016.
[13] *Tomb Raider*. Roar Uthaug. Warner Bros. Pictures, 2018.
[14] *Sonic the Hedgehog*. Directed by Jeff Fowler. Paramount Pictures, 2020.
[15] *Wreck-It Ralph*. Directed by Rich Moore. Walt Disney Studios Motion Pictures, 2012.
[16] *Ready Player One*. Directed by Steven Spielberg. Warner Bros. Pictures, 2018.
[17] *Citizen Kane*. Directed by Orson Welles. RKO Radio Pictures, 1941.
[18] *Vertigo*. Directed by Alfred Hitchcock. Paramount Pictures, 1958.
[19] *Touki Bouki*. Directed by Djibril Diop Mambéty. World Cinema Foundation, 1973.
[20] *An Elephant Sitting Still*. Directed by Hu Bo. Kim Stim (United States), 2018.
[21] *La Jetée*. Directed by Chris Marker. Argos Films, 1962.
[22] *Kitchen Sink*. Directed by Alison Maclean. Felix Media, 1989.
[23] *Jostle Bastard*. Developed by Pippin Barr. Pippin Barr, 2013.
[24] *A Series of Gunshots*. Developed by Pippin Barr. Pippin Barr, 2015.

as an experimental approach to finding and implementing more diverse ideas in game design, it struck me that this might also be an opportunity to continue my practice-based work on violence in games specifically. Cinema has long produced work about or including violence from a great diversity of perspectives. Just as adapting non-blockbuster cinema provides an opportunity to find design insight more generally, I believe it equally may suggest a more diverse and nuanced approach to the subject of violence.

The rest of this article is structured as follows. In Section 2, I discuss the methodological approach in detail, covering the formal design constraints in my adaptations, the selection of the 10 films chosen for adaptation, the technical platform used for implementation of the case study game, and the documentation process followed. Following this, in Section 3, I present the case study game itself, *Combat at the Movies*,[25] giving the reader an outline of the adaptations and their gameplay. In the discussion section which follows (Section 4), I identify and analyze three central themes that arose in the design and development of the game as reflected in the process documentation. Finally, in Section 5, I conclude by reviewing the overall trajectory of the paper and making some final remarks.

## 2. Method

The central premise of this article is that creating a series of highly constrained film adaptations alongside rigorous process documentation is a productive way to explore the hypothesis that film adaptation might be a method through which to uncover new game design ideas and insights. That is, while the methodology to be outlined below is highly restricted both in terms of cinematic inspiration and technical specifications, my eye is turned toward game design insight writ large rather than improving on or critiquing videogame film adaptation itself. The methodological elements described here include the formal constraints chosen (both in terms of game design and the choice of films), the technical platform used to create the work, and the documentation process employed.

### 2.1. Game Design Constraints

The key decision was to adapt films into the language and form of a *specific game*, creating a highly restricted set of rules to guide design. As already stated, this was in the interest of generating tension between film and design, but also provided for a more manageable design space to work within. The foundation game chosen, Atari's *Combat*[26] (Figure 1), an iconic work from the history of videogames, features two tanks on either side of a battlefield that must seek and destroy each other. The tanks can move and turn in 16 directions and can fire one bullet at a time at their opponent. Each time one tank destroys the other, play resets and the victorious player receives a point. Play lasts for a set time (two minutes and sixteen seconds) and the tank with the highest score wins. As with many Atari titles, the game also includes a sizeable number of variations of play via game modes, from different "mazes" to invisible tanks to bouncing bullets. These variations nicely echo my objective of creating "cinematic variations" of the game.

---

[25] *Combat at the Movies*. Developed by Pippin Barr. Pippin Barr, 2020.
[26] *Combat*. Developed by Atari. Atari, 1977.

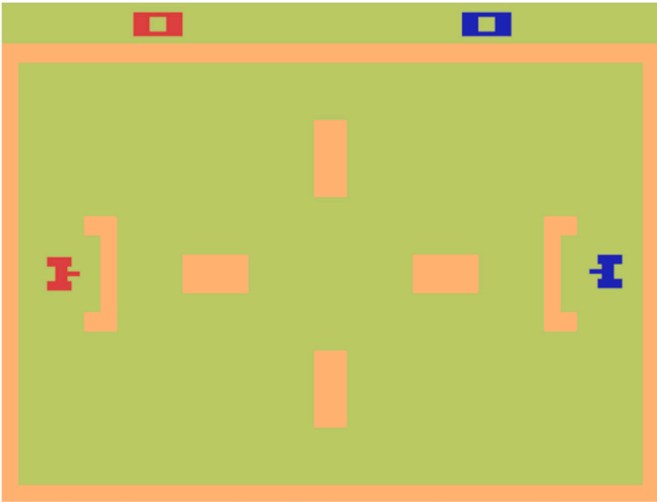

**Figure 1.** Atari's *Combat* (1977).

I selected *Combat* as the target for my adaptations for three key reasons. First, *Combat* is emblematic of numerous well-established ideas about game design, including balance and fairness, spatial exploration, skill-based play, and the centralizing of *combat* itself. Second, *Combat* (along with Atari games more generally) is a straightforward game in terms of its rules and representations, making it relatively more manageable to produce variations. Third, *Combat*'s distillation of videogame violence into a minimalist package struck me as a way to pursue my focus on the question of violence in videogame play through the movies adapted.

The choice to adapt *Combat* brings with it a host of constraints, most obviously in the form of the game's design and presentation, as above, but also from its underlying programming and the Atari hardware it runs on. While it was never my intention to dogmatically produce Atari-ready games of my own (by, for example, writing my adaptations in Atari's assembly language to run on the original hardware), I did want to leverage the restrictive nature of software and hardware to embrace both my central premise of design constraints, and to pursue at least some level of authenticity, an idea I will discuss in more depth later.

### 2.2. Film Constraints

In order to examine the hypothesis that adaptations of films outside the action-oriented, blockbuster genre could lead to innovation and provocation in game design, the 10 films chosen for adaptation were all selected from *Sight & Sound* magazine's Critics' top 100 ( 2012) list. While any list of this nature is, of course, subjective, the British Film Institute, publisher of *Sight & Sound*, strikes me as a reasonable authority for identifying a specific kind of highly regarded film outside the blockbuster arena. I have seen each of the 10 films selected fairly recently (often more than once) and, unsurprisingly, each was selected for at least some perceived affordance for adaptation into the *Combat* framework.

The films selected were: *Citizen Kane*, *Rashomon*,[27] *Tokyo Story*,[28] *L'Avventura*,[29] *Au Hasard Balthazar*,[30] *2001: A Space Odyssey*,[31] *The Godfather*,[32] *The Conversation*,[33] *Taxi Driver*,[34] and *Beau Travail*.[35] These films span 60 years of the history of cinema, hail from Britain, Japan, Italy, France, and the United States, and explore diverse subject matters.

I note that owing to the short-form nature of the adaptations proposed, I began with the intention to restrict the adaptation itself generally to a single scene from the movie. Further, in keeping with my parallel investigation into videogame violence, I prioritized films and scenes within them that touched on violence or death in some way. This was in aid of producing adaptations that could then draw on these often nuanced or alternative treatments of violence in game design. Thus, for example, *Citizen Kane*'s "Rosebud" scene provides a reflection on dying alone, *Taxi Driver*'s mirror scene examines the psychological *precursor* to active violence, and *L'Avventura* deals with violence only in an implied and enigmatic way through a woman's disappearance.

### 2.3. Platform

Selecting a platform for game development is a decision influenced by multiple factors, including existing familiarity, the match between platform and game design, and the accessibility of the final product. Compromises must inevitably be made that influence everything from the moment to moment work of development to the resulting aesthetics and gameplay of the game itself.

In this project, the adaptations were written in JavaScript using the Phaser 3 game engine library and run in a standard web browser (though not on mobile devices). Phaser 3 is a contemporary 2D game engine that provides various affordances for game developers, including, for example, handling of sprites, physics, game cameras, sound processing, scene management, input, and more.

An alternative approach that would have also been a strong choice is batari Basic, a programming language created specifically to enable developers to write programs for the Atari 2600 system itself. Creating the adaptation in batari Basic, or indeed in the Atari 2600's assembly language, would have been a way to maintain a strong fidelity to the original *Combat* while adapting films into its form.

Returning to the key factors mentioned above, however, Phaser 3 seemed the better choice in total. Perhaps most importantly, it is a development tool I was already deeply familiar with and it allows me to distribute the resulting game as a simple webpage. As an experimental game designer, I generally find the *accessibility* of my work to be paramount, and the ability to invite players to experience the games via a simple link meets this goal far better than either requiring them to download an emulator or, even more extremely, for me to produce an Atari cartridge and distribute it. Phaser 3, although not as authentic as batari Basic, is a necessary compromise made in the interests of accessibility that, as I will discuss later, led to various challenges and imperfections that were often frustrating or illuminating or both.

### 2.4. Documentation

The final element of this project is the approach to documenting the design and development process as it occurred. To this end, I adopted the Model for Design Materialization and Analysis (MDMA) documentation process created by myself along with Rilla Khaled and Jonathan Lessard (Khaled et al. 2018). MDMA is premised on the philosophy of the reflective practitioner (Schön 1983) along

---

[27] *Rashomon*. Directed by Akira Kurosawa. Daiei Film, 1950.

[28] *Tokyo Story*. Directed by Yasujirō Ozu. Shochiku, 1953.

[29] *L'Avventura*. Directed by Michelangelo Antonioni. Cino Del Duca, 1960.

[30] *Au Hasard Balthazar*. Directed by Robert Bresson. Cinema Ventures, 1966.

[31] *2001: A Space Odyssey*. Directed by Stanley Kubrick. Metro-Goldwyn-Mayer, 1968.

[32] *The Godfather*. Directed by Francis Ford Coppola. Paramount Pictures, 1972.

[33] *The Conversation*. Directed by Francis Ford Coppola. Paramount Pictures, 1974.

[34] *Taxi Driver*. Directed by Martin Scorsese. Columbia Pictures, 1976.

[35] *Beau Travail*. Directed by Claire Denis. Pyramide Distribution, 1999.

with approaches such as research through design (Godin and Zahedi 2014) and others. This process centralizes a cumulative, detail-oriented tracking of the design process by tying it to a code repository managed by Git. While code repositories are ordinarily used for maintaining a chronological and step-by-step account of how the code of a program is built up and changed over time, MDMA extends on this idea to include design thinking as another key element to be traced explicitly through the history of a project.

Throughout my development of the adaptations, I followed three key documentation practices. First, I wrote regular process journal entries which were stored in the code repository of the project itself. These entries function as high-level reflections on the design work being done and encompass thinking about everything from the objectives of the game overall (in terms of an investigation into film adaptation) to the specifics of particular elements of the development process (such as the Phaser 3 camera system). Because of the historical nature of the repository, each diary entry can be easily paired with a playable version of the game as it was in the development cycle at that precise moment, meaning we can see both design thinking and design outcome side by side.

The second documentation practice was to write short design messages with each addition of code to the repository. That is, once a specific chunk of programming (or other development) work was completed, I would formally add this new material to the repository and write a reflective note on precisely the implications of that technical work relative to the design objectives and process being followed. These shorter messages provide a chronology of design thinking as it pertains directly to the code and other materials from which the game is constructed (Figure 2).

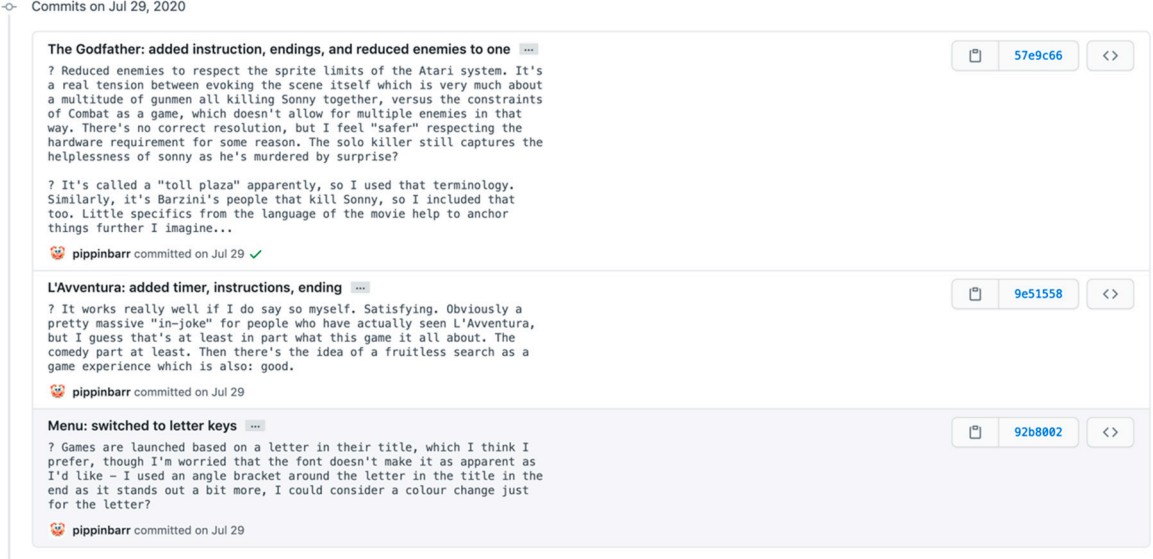

**Figure 2.** Design reflections in the code repository.

The final documentation practice, carried out much more occasionally, was to write entries I called "why" documentation in which I very specifically attempted to frame my argument about the research objectives of the project overall. These meditations, written at the beginning and then during the design and development process, allowed me to repeatedly return to the research questions at the heart of the project, helping to avoid the common pitfall of becoming lost in the maze of development rather than maintaining a focus on the research rationale for the project itself.

## 3. Case Study: Combat at the Movies

The game produced according to the methodology discussed in the previous section is titled, rather literally, *Combat at the Movies*. It is presented as a set of Atari games based on *Combat* that adapt each of the 10 selected films. I will here outline the presentation form of the game, the nature of the 10

adaptations within it, and the design objectives for each. Note that the games themselves as well as the full documentation of the design and development process can be found at the links provided in the Supplemental Materials at the end of this article.

### 3.1. Presentation

On beginning the game, the player is presented with a menu system based on the instructions manual for the original *Combat* (Figure 3a). This allowed me to provide additional information about the nature of the game. On selecting a specific film from the list, the player is then presented with a further instructions screen (again based on the instruction booklet) which explains the aesthetic and expressive objectives of the game, framed in language echoing the original instructions. Thus, for example, the instructions for *Citizen Kane* (Figure 3b) are as follows:

> CITIZEN KANE is a game of memories. Play Charles Foster Kane as he lies dying in Xanadu. Use the Left and Right Arrow Keys to toss and turn in your bed, filled with regret. Press the Space Bar to give voice to your one most precious memory. But better remember it quickly! You're going to die!

Each instruction screen includes a screenshot of the game paired with an image from the original movie chosen to show aesthetic and other comparisons between the two. These detailed instruction screens allow a significant amount of preparatory work to be done for the player so that they are more able to engage with the resulting adaptations, which are, admittedly, often strange and counterintuitive.

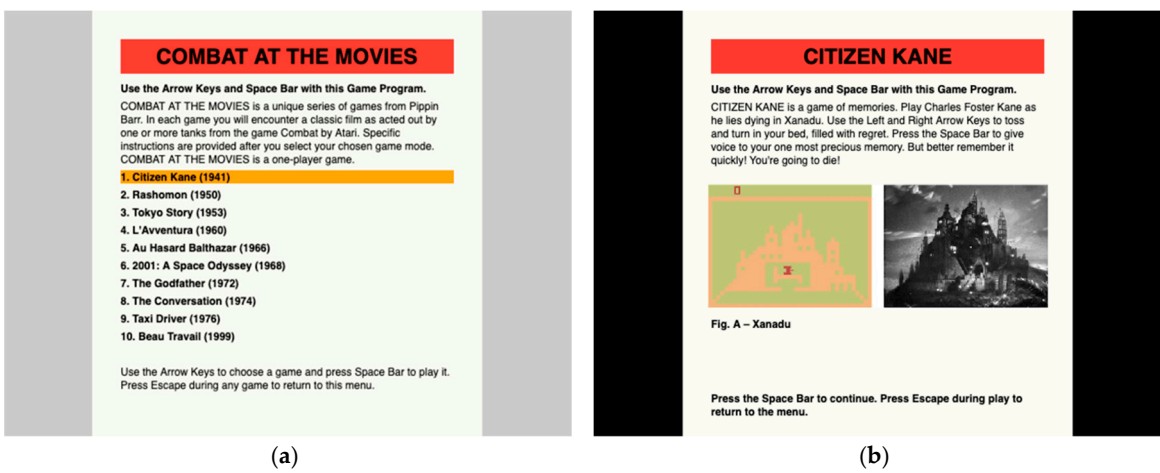

(**a**)          (**b**)

**Figure 3.** Instruction screens for *Combat at the Movies*. (**a**) The main menu with general description of the project; (**b**) the specific instructions for the *Citizen Kane* adaptation.

### 3.2. Game Descriptions

#### 3.2.1. Citizen Kane (1941)

This game reproduces the iconic death scene in the movie, focusing on the explicit act of memory by Kane just before he dies. Here, we see the standard *Combat* tank at rest on a bed inside a representation of Xanadu, Kane's castle-like home. The tank's usual "shooting" action is replaced by the tank whispering the word "Rosebud" repeatedly, rendered in a mechanical buzz based on the little-used voice synthesis on the Atari. The tank dies (represented by its characteristic spinning motion from *Combat*) after 20 s and the player scores points with each remembering of Rosebud. In order to offer some minimal gameplay, the player can also rotate the tank in place as a way of "tossing and turning" with angst.

### 3.2.2. *Rashomon* (1950)

In this adaptation, the player participates in the dueling scene from the film. This is one of the key examples of the unreliable narrators in Rashomon as the two key retellings (by the bandit and the samurai) are radically different. Rather than create multiple, distinct versions of the duel in game form, the adaptation shifts the idea of unreliable narration to a metaphor of four multiple, distinctly behaving cameras (echoing the four perspectives from the film). Each camera presents a portion of the standard *Combat* battlefield in a different way and out of order, with one making the maze invisible, another omitting the player's enemy, and another omitting the player themselves. Combat takes place as usual, with the player or the enemy tank emerging victorious amidst the confusion.

### 3.2.3. *Tokyo Story* (1958)

Yasujirō Ozu's classic meditation on family bonds ends with the distressing scene of Shūkichi Hirayama, now a widower after the death of his wife Tomi, sitting alone at home with nobody left to care for him. The game adaptation takes on this final image, positioning the player's tank in the standard *Combat* field of play (the "home" of the tank), but with no other tank to fight with. Mirroring the loneliness of Shūkichi, the player's tank, unmoored from its purpose, can only drive slowly through the battlefield, feeling its age, and perhaps even pining for the memory of its traditional enemy, the blue tank.

### 3.2.4. *L'Avventura* (1960)

The island of Lisca Bianca takes the place of the battlefield in this adaptation, with the game focusing on Sandro's fruitless search for Anna when she goes missing. In keeping with the nature of the film, the player's tank is alone in an empty outline of the island and can only drive around, bumping into the coastline, but never finding the elusive Anna. As there is no point to shooting in this situation, nor was any suitable metaphor or other use of the shooting mechanic found, it is disabled, leaving the player to wander until their time runs out. With respect to the investigation of forms of violence, I interpreted Anna's disappearance as a highly ambiguous form of violence in the film.

### 3.2.5. *Au Hasard Balthazar* (1966)

*Au Hasard Balthazar* shows us the odyssey of the titular donkey and his (often cruel) intersections with humans. Rather than recreating one specific scene, the adaptation takes this higher-level idea of the movie and presents the player with a standard *Combat* battlefield, complete with maze elements, but with a donkey (Balthazar) in place of the enemy tank. True to the spirit of the movie, Balthazar wanders the screen at random and does not engage with the player. The player, representing humanity, can leave the donkey alone or pursue and kill him. Whatever is decided, as in the movie, Balthazar dies in the end, either at the player's hands or at the end of a timer representing his lifespan.

### 3.2.6. *2001: A Space Odyssey* (1968)

The most mechanically complex of the adaptations, this game takes as its subject the early scene in which a group of apes gains an evolutionary edge by learning how to use weapons from an imposing black monolith. In the game, we see a standard empty playing field with the player and enemy tank in their starting positions. On play beginning, neither the player nor the enemy is able to shoot, in keeping with the movie's narrative. After a short period of time, the monolith appears on the player's side of the field. Bumping into it causes it to disappear and the player tank (but not the enemy tank) to gain the ability to shoot. The "evolved" tank is then free to hunt down and eliminate the enemy in a completely one-sided fight.

### 3.2.7. *The Godfather* (1972)

Addressing the famous toll plaza scene, this adaptation positions the player as the helpless Sonny Corleone, doomed to drive up to the toll station only to be murdered by the Barzini gang in an ambush. The game version reproduces the scene quite literally in the visual and interactive language of *Combat*, with the playing field representing a road leading to a toll station, and the enemy tank suddenly appearing and shooting the player when they reach a specific point along the road. As in the movie, the player's tank cannot defend itself in time, leading to an experience of helplessness and an "unfair" death.

### 3.2.8. *The Conversation* (1974)

In the *Combat* version of *The Conversation*, the player takes on the role of Harry Caul, a surveillance expert tracking a potential murder plot. The game reproduces the famous hotel scene toward the end of the film, in which Harry listens through his room's wall to a fight that ends in murder. Keeping the focus on the auditory elements, the game makes the contents of the playing field invisible, even as two game-controlled tanks move within it and, eventually, fight to the death. The player is only able to hear the sounds of the combat and is left unsure of the result, as in the movie.

### 3.2.9. *Taxi Driver* (1976)

In this game, the player participates in the iconic mirror scene in which Travis Bickle confronts his own reflection and practices his tough-guy routine. He draws his hidden pistol and delivers the well-known line, "you talkin' to me?" In the game adaptation, the player's tank is Bickle, positioned in front of a mirror in an otherwise empty playing field. The player can reposition their tank and move around, but must be visible in and facing the mirror in order to use the "shoot" button. This triggers a heavily distorted (in keeping with Atari speech synthesis) version of the line. The player receives a point each time they successfully threaten themselves in the mirror until time runs out.

### 3.2.10. *Beau Travail* (1999)

The beautiful final scene of the movie, in which Denis Lavant as Chief Adjutant Galoup dances alone in front of a sparkling mirrored wall to Corona's Rhythm of the Night, is the subject of the final game. The player's tank is in front of a duplicate mirror wall, complete with an imitation of blinking lights, and can express themselves through dancing movements of their vehicle until a timer ends. As a concession to the expressive nature of movement, this is the one game in which the tank can move backwards as well as forwards, allowing for more creative arabesques and swoops.

## 4. Analysis

The process of designing, developing, and documenting 10 film adaptations into the form of Atari's game *Combat* led to a significant amount of data. Foremost, are the games themselves, playable online, each reifying the overall project's objectives to explore the potentials of film adaptation in a highly restrictive design context. Alongside this, the documentation of the project yielded over 10,000 words of process journal entries, over 5000 words of detailed technical commit messages, over 1500 words of "to do" tasks and completion notes, and over 1300 words of high-level design research discussion in the "why" document. All this data is publicly available in the game's Git repository at https://www.github.com/pippinbarr/combat-at-the-movies.

In reviewing the documentation, three key themes were identified as particularly worth discussing in the context of this exploration of film adaptation: *adaptation in code*, *prioritizing Combat*, and *reframing violence*. In the following subsections, I discuss these themes in detail alongside quotes from the process documentation and examples from the games themselves (including older versions). Seen together, these themes help us to reach an understanding of *Combat at the Movies* as a research design object that

raises questions and points in potentially generative directions for film adaptation as an experimental videogame method.

### 4.1. Adaptation in Code

Beginning closest to the materials of game creation, it became clear that we can and should consider in some depth the direct relevance of the code itself when thinking about questions of adaptation. In terms of the specific adaptations pursued here, a recurring theme revolved around correspondences between film and underlying game code. This sometimes involved making sure that implementation details were true to the film being adapted. Thus, in *The Conversation*, rather than variables (names for data in the program) representing the "red tank" and "blue tank", the two tanks which fight it out in the unseen hotel room are named "husband" and "wife" in keeping with the film's plot (Figure 4). While such a low-level change has no bearing on the player's experience, it both supported and further provoked a level of technical thoughtfulness in my design and development activities. Knowing that the code could (and perhaps should) be in sync with the surface-level representations led to my reflecting significantly more deeply on these potential correspondences and contradictions elsewhere.

```
76          this.husband.update();
77          this.wife.update();
```

**Figure 4.** Husband and wife in code.

Indeed, at other times, conventional game programming naming conventions served in part to *interpret* the film itself, such as in the case of *Taxi Driver*. Here, the effect of the player being able to see their tank in a mirror is produced by generating a second tank which mirrors the movements of the player's tank. If I were pursuing a similar strategy of matching implementation to the truth of the movie, I might have called this second tank "mirror image" or something similar. In this case, however, the code uses the default variable name for the second tank that I had been using from the beginning of development: "enemy". Thinking simply about a game of *Combat*, this makes sense as a name for the "other" tank that the player seeks to destroy, but for a tank used as a mirror image of the player, it serves further as an (unseen) intensifier of the scene itself: the mirror image really is Travis Bickle's enemy, both because he is pretending to confront an unnamed stranger with his gun, but also in the sense that Bickle is his own antagonist here. Although a detail such as this is "buried" within the code itself, it does lend itself to an argument that the very *implementation* of film adaptation can serve as a form of film criticism and insight.

There are numerous other examples of these kinds of affinities and commentaries between code and film at work. The use of multiple "cameras", a specific affordance within the Phaser 3 library for providing distinct views of a scene, to serve as a metaphor for the multiple perspectives in *Rashomon* indicates that implementation details can move beyond literal translations to the figurative. In the opposite direction, the fact that the implementation of the donkey Balthazar's movement through space in *Au Hasard Balthazar* is a function called "randomMovement" provides a very literal correspondence between the "hasard" (chance) of the title and the behavior of its star (Figure 5). Finally, in *L'Avventura* there is the existential question of whether there should be, secretly and invisibly, a tank to represent the missing Anna in the film. To include this "Anna tank" in the underlying code would be to say that she is *somewhere*, and simply cannot be found by the player. In the end, I felt this worked against the mysterious disappearance the film centers on, preferring for her not only to have vanished from the low-resolution island in the game, but from the very source code itself (Figure 6).

```
40    randomMovement() {
41      let turn = Math.random();
42      if (turn < 0.01) {
43        this.rotationDirection = -1;
44      }
45      else if (turn < 0.02) {
46        this.rotationDirection = 1;
47      }
48      else {
49        this.rotationDirection = 0;
50      }
51
52      this.speed = Math.random() < 0.1 ? 0 : this.maxSpeed;
53    }
```

**Figure 5.** Balthazar's random movement.

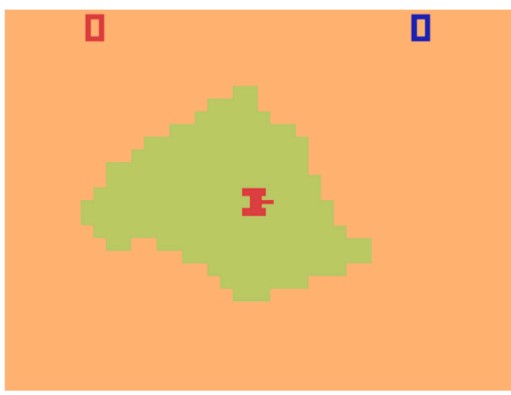

**Figure 6.** Not only is no tank representing the missing Anna visible here, there is no Anna in the source code itself.

Again, while a player of *Combat at the Movies* will not be able to detect the above and other specifics of code, I believe there is a special beauty and authenticity to be found in tracking the adaptation process all the way to implementation details. Indeed, the discussion here reinforces that, in keeping with platform studies (Bogost and Montfort 2009) and software studies (Marino 2020), the code level is absolutely part of any complete analysis of a videogame. Furthermore, as indicated above, it seems clear that an adherence to this level of commitment could be fruitful both as a kind of game design film criticism (as with *Taxi Driver*), as a way to maintain designer accountability to the source material (as with the attempt to be true to *L'Avventura* by maintaining the mystery of Anna's disappearance in code), and to follow design trajectories that might otherwise go unexplored.

### 4.2. Prioritizing Combat

In any process of adaptation, there is a need to determine our priorities: when they come into conflict, should we prioritize the source material or the medium we are adapting to? These moments of tension between film and videogame characterized many of the most generative design moments while working on *Combat at the Movies*. As I worked, I found my way toward the principle that, generally speaking, I would privilege the target game form of *Combat* (and its associated software, hardware, and design limits) and attempt to find the best *fit* for the films I was adapting. This is very much a personal decision in keeping with my own practice of experimental videogame design. I have long privileged formal constraints of this nature in other projects, such as my translation of philosophical isms into the game form of the classic Snake game in *SNAKISMS*,[36] or my adaptation of

---

[36]  *SNAKISMS*. Developed by Pippin Barr. Pippin Barr, 2017.

Greek mythologies of punishment to the platform of Windows 95-era user interface elements in *Let's Play: Ancient Greek Punishment: UI Edition*.[37]

As an example of formal limits posed by a target platform/game, consider that the Atari 2600 placed a precise limit on the number of sprites (dynamic visual elements) that could be displayed. Although this limit was exceeded by some games through pleasing tricks, I chose to stick with the key limitation of only two tank sprites. This core limitation is in no way enforced by the Phaser 3 library used to implement *Combat at the Movies*, which can support more or less arbitrary numbers of sprites on screen at the same time. Rather, I embraced the decision both as a form of authenticity, making the resulting game more recognizably in the style of Atari (and *Combat* specifically), as well as a way to find generative constraints for design.

The sprite limit led immediately to key decisions in at least two of the adaptations. In *The Conversation*, in which the player listens to an unseen fight between two tanks representing the husband and wife from the movie, I had included a tank for the player to control *outside* the battlefield, as if they were outside the wall of the hotel room. I later realized, however, that although the two tanks inside the hotel room were invisible to the player, they still necessarily counted as sprites in the game (having positions, collision detection, and more), placing the player's tank in contravention of the sprite limit, as I noted in a commit message on the subject (Figure 7). I removed the player's tank and the game became one purely of listening, without the distraction of moving a tank around—in many ways very much more in keeping with its cinematic source.

```
? Realized that a) having the player sprite plus two enemy sprites
breaks Atari sprite rules, and b) although there's a player tank
everywhere else, all the player actually does in this game is listen, so
the tank is extraneous (completely obvious when you consider I placed it
outside the playing field). With it gone the game is much more pure and
accurate.
```

**Figure 7.** Design documentation from the moment of removing the player tank in *The Conversation*. Commit c993d80.

In *The Godfather*, the limitation had a more diminishing effect on the recreation of the toll plaza scene. Whereas in the film, Sonny Corleone is attacked by a multitude of Barzini's men from all sides, the sprite limit of *Combat* requires that the player tank be attacked by exactly one other tank. As with *The Conversation*, I initially implemented the film with eight enemy tanks bearing down on the player (Figure 8a) before recognizing it was not in keeping with *Combat*. Thus, in the new version, the player drives through the plaza and is met with an ambush from just one other tank (Figure 8b). The success or failure of this design decision in play rests on whether a player feels a greater loss at the ambush not being fully represented, or a greater sense of coherence at the basic structure of *Combat* being maintained. Whatever the case, it is illustrative of a commitment to a specific design trajectory and how it interacts with the activity of adaptation.

---

[37] *Let's Play: Ancient Greek Punishment: UI Edition*. Developed by Pippin Barr. Pippin Barr, 2019.

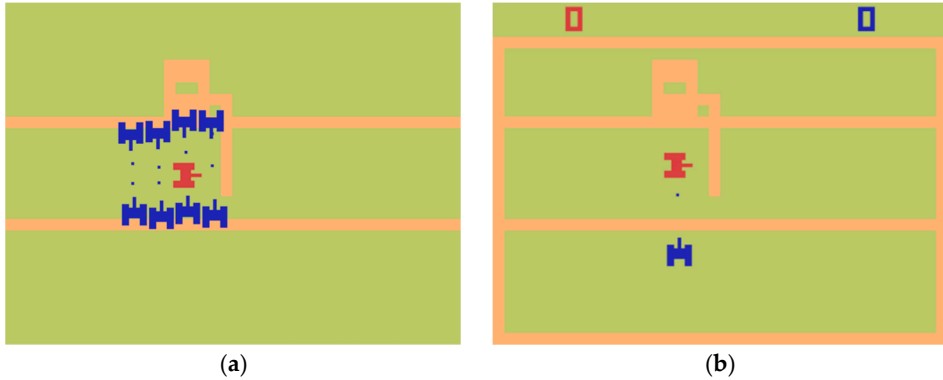

| (**a**) | (**b**) |

**Figure 8.** The ambush in *The Godfather*. (**a**) As originally envisaged with multiple attacking tanks in keeping with the film; (**b**) as reconceived with a single "Barzini tank" to adhere to *Combat*'s platform restrictions.

A further source of generative tension resides at the level of the *mechanics* of *Combat*. As discussed earlier, *Combat* is a game with highly restrictive possibilities for player agency and outcomes, which boil down to driving and shooting and dying. All adaptations from the films chosen had to fit more or less within these basic forms of agency in the blocky *Combat* world. Thus, movement most often replicates the idea of traversing space toward some Other, but is also used to represent searching in *L'Avventura*, dancing in *Beau Travail*, and tossing and turning in bed in *Citizen Kane*. Shooting is "just shooting" in *Au Hasard Balthazar* and a stand-in for sword fighting in *Rashomon*, but also a trigger to "fire" out the line "you talkin' to me?" in *Taxi Driver* and is completely absent in *L'Avventura*. Death, as will be discussed in depth below, takes on multiple forms too.

Beyond maintaining a fairly rigid adherence to the design norms of *Combat*, there are also places in which a little more poetic license has been allowed to slip into the implementations of the films. In *Tokyo Story*, where the player's tank exists alone in the world, the movement and turning speeds have been adjusted downwards to emphasize the idea that the tank (in the role of Shūkichi Hirayama) is elderly (Figure 9). In *Beau Travail*, the ability to reverse has been added to the tank controls in order to expand the expressive language of movement available to the player in keeping with the beauty of the dancing in the movie.

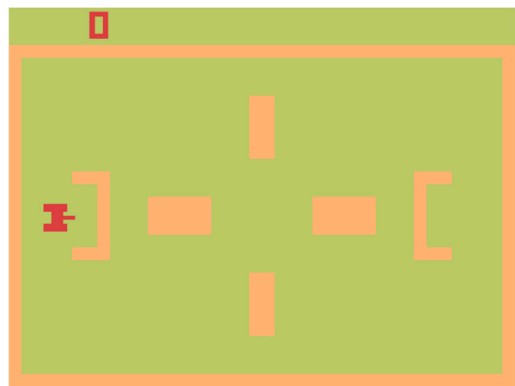

**Figure 9.** The player is alone in the world of *Tokyo Story*, with no family to turn to and likewise no enemy to pursue.

In a way, the relationship between the selected films and the established framework of *Combat* serves as an illustration of the nature of adaptation itself: a series of pointed compromises between the source and the new work. In order to "pick a side" at each moment of tension, the designer must think explicitly and in some depth about both sides of the equation, suggesting this form of adaptation

may be a particularly useful exercise in videogame design thinking as much as a means to create a final product.

### 4.3. Reframing Violence

As introduced in Section 2.1, this project was intended not only to explore the adaptation of non-blockbuster films into a constrained videogame language, but also to use this process as an opportunity to comment on videogame violence. The games in *Combat at the Movies* each involve a direct or implied relationship to death and/or violence that complicates or outright contradicts standard videogame interpretations of these ideas. Each game presents, thanks to its cinematic source, a different way of thinking about what it means to kill, to be killed, to die, or to act violently. While I will not discuss every game here, I will provide three examples to clarify what I mean by this.

A useful place to start is with *Taxi Driver*. Here, as in many games, the player assumes the role of a violent character, Travis Bickle. Bickle is in some ways a lot like the *Combat* tank, bent toward destruction and death, but here we find Bickle at home in front of his mirror rather than in an arena for battle. In the game, the player confronts their reflection in the mirror and speaks the famous line, "you talkin' to me?" (Figure 10). The violence is detuned in the literal sense that no shots are fired and nobody dies. Instead, the player goes through the motions of threatening an imagined adversary. This idea of a precursor to violence, of an antagonist working themselves up into the emotional state needed to commit murder, is rarely thought of or included in game designs. Taken seriously, it draws the violence of the tank outside the cycle of kill-and-be-killed in a way that may make it at least somewhat affectively accessible. In a sense, the *Taxi Driver* adaptation could be read not just as an adaptation of a movie, but almost a vision of an imaginary "locker room" in which the original *Combat* tank psyches itself up for battle.

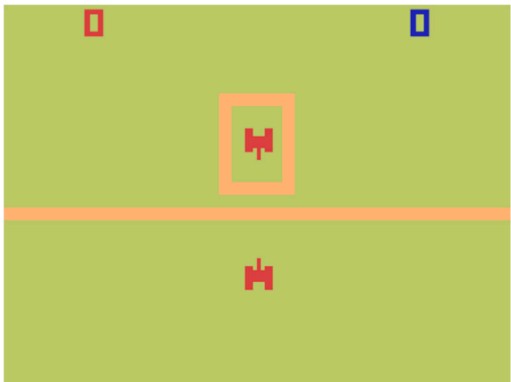

**Figure 10.** Travis Bickle, the tank, confronts himself in the mirror.

More straightforwardly, *Au Hasard Balthazar* presents the player with a simple choice: Balthazar, the donkey, wanders the screen aimlessly and the player can kill him or not (Figure 11). The donkey is of no threat and can be killed without any official negative consequence, such as loss of points. Despite the fairly obvious *moral* answer to this "dilemma" (leave the poor donkey alone), many players may feel that killing Balthazar is the implied goal, or at least desirable, because it is the most *active* way to play. This premise in videogame play that if one *can* do something (such as kill) then one *should*, or at the very least should *try it out*, is at the heart of many internal justifications for "unnecessary" violent acts in play. This essential cruelty that players exhibit toward virtual characters, killing or wounding them because they can, is very much in keeping with the arbitrary and needless cruelty of many of the people we meet in *Au Hasard Balthazar* the film. Whether or not a player of the game feels even a modicum of remorse on killing the innocent donkey (or conversely any satisfaction at leaving him alone), they take part in a videogame mirror of the world Bresson presented to us.

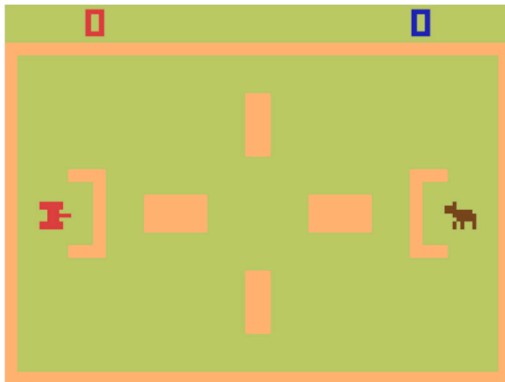

**Figure 11.** The player shares the battlefield with a harmless donkey.

To choose one final adaptation to illustrate these ideas, consider *The Conversation*. Here, as already stated, the player is not the enactor of violence, and indeed does not even have a controllable representation in the world of the game. Rather, they *listen* to the violence of invisible others and, per the plot of the movie itself, seek to discern what is happening behind closed doors. Bearing witness to violence and death rather than being its agent is another underused approach to violence in the context of videogames, where players are ordinarily placed in positions to act violently toward others for various reasons, such as war (e.g., *Call of Duty*[38]), saving damsels in distress (e.g., *Super Mario Bros.*[39]), or simple criminal cruelty (e.g., the *Grand Theft Auto* series[40]). In removing the player's ability to act decisively with violence but maintaining a violent world, the player of *The Conversation* is asked to bear witness instead of jumping in themselves. While there are games that use this approach (most notably *Return of the Obra Dinn*[41]), it strikes me as a significantly powerful direction of investigation for game designers interested in engaging with violence as a human concern.

Without going into the knotty debate on the impact of violence in games, we can at least agree that the treatment of violence in games trends toward trivialization or, at best, a prioritization of fun and tests of skill. Adapting non-action-oriented films, all of which still intersect with an idea of violence or death in some way, gave me a chance to consider a diverse set of alternate approaches, as illustrated above. In practice, this exploration led to the idea that film adaptation (outside the world of blockbusters) could be a generative source of other understandings of violence and that the specific design results could be used as principles in games which are not explicitly adaptations, much in the way they have borrowed from films' visual language.

## 5. Conclusions

In this article, I have made a case for film adaptation as a worthwhile experimental design approach. I began with a single research question: *what potential game-design insights can be gained from adapting unconventional cinematic choices?* Methodologically, my strategy has been to pursue a practical engagement with this question by actively *creating* such unconventional film adaptations, with the additional structuring principle of also adhering to a particular game form (Atari's *Combat*). Thematically, it has also been my particular aim to grapple with alternative representations of violence in play by relying on the nuance and diversity of the cinematic portrayals selected. As such, the core of my answer to the question is presented in the form of the game *Combat at the Movies* itself, with players of that game in a position to judge whether it achieves the aim of finding novel design approaches through

---

38    *Call of Duty*. Developed by Infinity Ward. Activision, 2003.
39    *Super Mario Bros.* Developed by Nintendo. Nintendo, 1985.
40    *Grand Theft Auto Series*. Developed by Rockstar Games. Rockstar Games, 1997–2013.
41    *Return of the Obra Dinn*. Developed by Lucas Pope. 3909, 2018.

the lens of adaptation. For myself, it has been clear throughout the project that this adaptation-focused design approach can be a wellspring of *new* design-thinking, stimulated by both the disjunctions and affinities of cinema and videogame design, with novel treatments of violence being a particular success here.

Beyond the game itself, the research at hand takes the form of reflective consideration and detailed analysis of the *process* of design in this specific context of adaptation. In the course of my writing, as well as in the process documentation at large, I hope it is clear that this design approach is highly generative for design thinking, provoking a well-defined series of decision points that must be addressed intelligently and with reference both to source material and target game form. I even suggest that through this reflective practice, those who address adaptation from other perspectives (from practitioners in other media, such as literature of film, to theorists of adaptation) may find material of interest in this highly specific project. It may be that seeing adaptation reflected in the mirror of videogame design and development can shed light on important aspects of adaptation itself, from the question of interactivity to the potential role of computer code as a medium.

Outside its clear value for thoughtful designers who are looking for a challenge, this quality of emphasizing *design itself* suggests to me a pedagogical role for film adaptation in game design education, with the highly structured task pressing students at all levels to grapple with specific design problems. Indeed, I note that the structured approach to design presented here is easily replicable. A literal adherence to the approach presented here would lead the willing designer to select one or more films and an existing game format and to jump immediately into the generative process of design. It is also more than possible to take on the challenge posed here in other forms, perhaps by relaxing some of the more intensive restrictions, such as that of authenticity to a platform such as the Atari 2600. Whatever the case, I hope that these ideas find their way into the hands of designers with many perspectives who might join in the highly enjoyable task of film adaptation.

**Supplementary Materials:** The full project documentation for *Combat at the Movies* is available at github.com/pippinbarr/combat-at-the-movies. The game itself can be played at pippinbarr.github.io/combat-at-the-movies.

**Funding:** This research received no external funding.

**Acknowledgments:** The author would like to thank Rilla Khaled, Jim Barr, and Mary Barr for extensive discussions both of this article as it was being written and the game *Combat at the Movies* itself during design and development.

**Conflicts of Interest:** The author declares no conflict of interest.

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
