# Peer review of "Film Adaptation as Experimental Game Design"

_arts_

Round 1
Reviewer 1 Report
This is a very convincing paper, that questions the function and affordances of video games as a medium for adaptation of film content. The proposed methodology (research through design) is suited to the task, and the result of the research (the game Combat at the movies) is a testament to an insightful critical process.
I think the author may want to look into two thing I found lacking.
1) I think the paper needs a larger discussion of why the author decided to use Phaser 3 as a programming language instead of using tools such as BAtari, which would have allowed the author to work with the actual limitations of the VCS instead of self-imposed ones. I can see a number of reasons why this choice was made, but the discussion in the paper is underdeveloped at the moment.
2) The section on film/video game violence feels a little disconnected from the rest of the paper at the moment. For example, I wonder whether the set of films chosen by the author is a good fit for both research questions (how can we adapt non-blockbusters? and how can we represent film violence in games in non trivial ways?), or if, rather, two different sets of films would have been more suited. Also, I think that the author should make an effort to clarify why the question of violence (as opposed to love, conflict, friendship, etc.) is relevant here. At the moment the author puts it as a sort of follow up to a previous study, but this is not enough to justify its presence in the paper. I think the section on violence needs some work in order to be consistent with the rest of an otherwise excellent work.
Minor spelling issues/typos: line 33: "In some" should read "In sum".
line 39: there should be a comma after "films".
Author Response
I would first of all like to register my gratitude to the reviewer, whose comments have, I believe, greatly strengthened the article.
I think the paper needs a larger discussion of why the author decided to use Phaser 3 as a programming language instead of using tools such as BAtari, which would have allowed the author to work with the actual limitations of the VCS instead of self-imposed ones. I can see a number of reasons why this choice was made, but the discussion in the paper is underdeveloped at the moment.
This is a hugely useful comment, and one I take very much to heart. I agree I was not specific enough in my discussion of platforms, perhaps seeking to avoid engaging with technical subjects in an article for a potentially non-technical audience. Nonetheless, it is a key part of the equation here, and I have added a substantial amount of discussion to the paper in section 2.3 (beginning on line 63) to explicitly address the factors and compromises being made in selecting Phaser 3 as my platform as opposed to working with a language such as batari Basic that could literally produce Atari-compatible software.
The section on film/video game violence feels a little disconnected from the rest of the paper at the moment. For example, I wonder whether the set of films chosen by the author is a good fit for both research questions (how can we adapt non-blockbusters? and how can we represent film violence in games in non trivial ways?), or if, rather, two different sets of films would have been more suited. Also, I think that the author should make an effort to clarify why the question of violence (as opposed to love, conflict, friendship, etc.) is relevant here. At the moment the author puts it as a sort of follow up to a previous study, but this is not enough to justify its presence in the paper. I think the section on violence needs some work in order to be consistent with the rest of an otherwise excellent work.
I appreciate this critique, and it is indeed something I had been thinking about during the writing of the article across its various drafts. While I have tried to be clear about the theme of violence that is a part of the project, I agree I can do better.
In response, I have added to the paragraph on violence in games (beginning on line 63) a fuller discussion of my rationale for thinking that adaptation might provide a worthwhile path toward contemplating violence in games.
Further to this, I have expanded the discussion in section 2.2 (beginning on line 155) on film constraints in relation to those selected films' relationships to violence as a contributing factor to their selection. In particular, I have not just stated that the films relate to violence, but have provided examples that may help the reader better understand the specific selection of films.
line 33: "In some" should read "In sum".
Done.
line 39: there should be a comma after "films".
Done.
Reviewer 2 Report
You put so much limitations in your design process, that you end up presenting a work that has historical significance, as something more universal than what it actually is.
reading your article I understood a lot of the difficulties Game developers could be facing when being tasked to adapt movies at the time of the Atari. But are your experiments still sound to draw conclusions about modern movie adaptations, I am not so sure. You spent a lot of time in the article explaining that you tried to stick as much as possible to the limitations of « Combat, » but I felt like keeping those limitations when they don’t actually help you answering your initial research question was going a bit too far.
Maybe a little reframing of the new knowledge that your experiment brings would make the article more relevant.
Anyway, there is a lot of work that went though this research so I would like to congratulate you on the good job anyway!
very good that the whole research process is documented and accessible.
very good that you give an address to test the games for ourselves.
Your chapter was about code is maybe too long and does not give much insight to the average reader of Arts.
maybe a slight rewrite more focused on the philosophy that you applied in coding would greatly improve the article.
I think you should look a little more into references about « games as interactive digital narratives »
in some of your adaptations, giving points to the player for doing an action seemed a bit gratuitous and trying too hard to « gamify » an experience that could in the end just be « experiences. »
Best regards and good luck for the rest!
Author Response
I very much appreciate the feedback from the reviewer, they presented me with a perspective on my work that comes from outside my usual domain and this was very helpful not just in the specific critiques I address below, but more generally as I reread my work and considered its accessibility. Although much of the discussion that follows here does involve explanation/discussion rather than changes to the article, I hope the reviewer can see how much I have taken their input into serious consideration.
reading your article I understood a lot of the difficulties Game developers could be facing when being tasked to adapt movies at the time of the Atari. But are your experiments still sound to draw conclusions about modern movie adaptations, I am not so sure.
I suspect this is a case where I could be clearer about the specific intent behind this project, which is not so much to improve on contemporary videogame adaptations of films, but rather to use this general idea of adaptation as a lens through which to view and perhaps intervene in game design in general.
One culprit here may simply be a lack of clarity in the introductory section concerning my objective. Although I have already included language to the effect that my overall aim concern innovation in design writ large (seen through the lends of adaptation), I have inserted further emphasis of this both in the introduction (lines 49-50, lines 53-54) and in the methodology (from line 93).
Further to this, and in response to commentary from another reviewer, I have worked to improve the concluding section of the paper in order to re-emphasize the objectives set at the outset (from line 527).
You spent a lot of time in the article explaining that you tried to stick as much as possible to the limitations of « Combat, » but I felt like keeping those limitations when they don’t actually help you answering your initial research question was going a bit too far.
This is a reasonable observation, and I can understand how the adherence to limitations in the face of difficulty can appear to be needless. I do see the formal structure of the experiment (including the limitations framed by Combat) as central to the argument being made through practice. In particular, it is through engagement with limitations and the ways in which a designer responds to them (either literally in the design work itself, or reflectively in their discussion of it) that can shed light on a project. For me, these limitations, even when they may appear adverse to the research question, are crucial to the clarity and coherence of the experimental method.
Maybe a little reframing of the new knowledge that your experiment brings would make the article more relevant.
I appreciate this comment, and understand the desire for outward looking outcomes in the conclusion. I may have been overly tentative in this regard simply because of the singularity of the project discussed - it can feel like overreaching to suggest that there are substantial, actionable forms of knowledge that can come from a single engagement with adapation like this.
Nonetheless, as I mentioned in an earlier response, in the conclusion section I have attempt to clarify some of the key take-aways that I think are of value to a reader, particularly in the context of game design, but also more generally in relation to adaptation as a form of design.
Your chapter was about code is maybe too long and does not give much insight to the average reader of Arts. Maybe a slight rewrite more focused on the philosophy that you applied in coding would greatly improve the article.
Writing about the code level of a project is a significant challenge, and I accept that it may have felt inaccessible in places. I worked closely with readers who are not programmers (or generally technically inclined) while writing the article in a continuing attempt to clarify my language and ideas in this domain. I agree it is not perfect, but in the end see this as a much larger ongoing process to improve my ability to write about code for broad audiences. The suggestion of focusing on philosophy is noted, though in attempting to find a more general level of discourse, I worry that specificity, which I value highly, can often be lost.
I think you should look a little more into references about « games as interactive digital narratives »
I appreciate the point here. There is no question that there are significant domains of literature, including work on interactive narrative, that I have not pursued as part of this paper. I can only do so much with this specific piece of work, and have chosen to focus more or less exclusively on an exploration on the practicality and design knowledge involved in creating videogame adaptations. For future work, in which I certainly should begin to tie these practical insights and approaches more to existing theory and literature, I will be sure to include consideration of interactive narratives especially.
in some of your adaptations, giving points to the player for doing an action seemed a bit gratuitous and trying too hard to « gamify » an experience that could in the end just be « experiences. »
There is no question that there are alternative design decisions that could have been made at countless moments throughout this project. I appreciate your pointing to a specific moment that could have gone differently, though I assure you that my treatment of points was seriously considered and intended. I certainly don't disagree with the gratuitousness you note here and its connection to gamification - indeed I find those parallels generative and was interested in the tensions generated between the subtleties of cinematic language and the often crass nature of points in these adaptations. We may have to put this one down to artistic differences.
Reviewer 3 Report
The article ”Film adaptation as Experimental Game Design” explores to what extend the adaptation of what the author calls ‘unconventional cinematic’ choices can be used in a research-through-design approach to gain insights on game design and the representation of violence in games. Based on this, the article should be assessed based on its contributions to the two overall areas of research, namely (1) adaptation studies and in particular studies of game adaptations of film and (2) design research in game design.
As I see it, the article’s most significant contribution is in the area of research-through design, whereas its engagements with of the notion of adaptation remains somewhat superficial. The article introduces itself with a short discussion of game adaptations of film, which the author claims are characterized by a tendency to adapt blockbuster action movies that already fit established representational conventions of computer games and their portrayal of violence. By doing this, the author argues that game designers miss out on a potential of adapting a more diverse set of cinematic works. In addition to this, the author argues that looking beyond Blockbuster productions, and their cinematic devices and narrative structures will enable game designers to expand on the expressive capacity of the medium of games. As I read the first part of the this argument, the author sketches out a problem in the state of the art in film-to-game adaptations, which I believe is a result of the economic structures that both the AAA-game and the big-budget film industry is part of. But the author’s solution to this problem is much more formalist in nature exploring and expanding on the expressive potential of games. This then raises the question if such a formal inquiry really poses a solution to a problem that I believe is caused by more economic factors.
Regardless of this, I agree that drawing inspiration from a broader set of cinematic works is worthwhile and interesting – I see a potential in using this approach in educational programs in game design. In continuation of this, the authors framing of this project as research-through-design is a very good choice.
The author imposes two creative constraints for the project. First, the chosen films should be adapted not just to games, but to a specific game, namely the Atari classic Combat The reasons for this is that Combat exemplifies games in a ‘pure’ form (in my opinion a somewhat problematic statement), but also more pragmatically, that the game is relatively simple and straightforward in terms of it’s rules and representations. For an adaptation researcher this choice raises many interesting questions, such as whether this could then also be considered an adaptation of Combat (this is not unlike Randall (2017) who argues that the important ‘source’ even in game adaptations is the genre, and that the game’s relationship to its filmic or literary source should be considered paratextual). Other questions that could have been raised includes what it means to adapt to – and from – a game and what can be transferred from games (only the ‘contents’ of the text, such as plot, characters and objects, or also their ‘formal’ aspects such as rules, mechanics, engines and so forth. Furthermore, basing a game on an existing, specific and named rule-set (D&D in table-top roleplaying games) or an engine (such as Valve’s Source) is common in game design but is not framed as adaptations, but should it be, and what are the implications of such as framing? Such considerations are unfortunately lacking, and the author, with a few exceptions, does not engage with any existing literature on adaptations (such as Hutcheon 2006; Bruhn et al. 2013; Leitch 2008, 2017,) or specifically about game adaptations (such as Picard 2007, Randall 2016, Flanagan 2017, Svelch 2020 – nor to the many the many works on transmediality such as Tosca and Klastrup 2004, 2014, 2016, Thon and Ryan 2014). This is a shame, because I believe this would also inform the analysis of the author’s own games and their design process. But it is perhaps fair enough, since the author’s focus is clearly on game design (using adaptation mostly as an inspiration or starting point), and not on theorizing (game) adaptations as such, which is clear from both title and contents of the article. Still, I suggest the author to at least clearly delimit the current practice-based inquiry into game design with these more theoretical or analytical contributions.
The second constraint of the project is, that the practice of adapting the films to games should reflect considerations of the representation of violence and aim at exploring new ways of treating the subject of violence in games. This constraint is imposed on the choice of filmic scenes to adapt, that the author argues in some way touches upon death and violence. The filmic scenes themselves are left relatively undescribed, and only through descriptions of the game adaptation will the reader get insights into the source scenes, and the authors interpretation on them. I think this is telling for the article in its entirety: the focus is clearly on the games, and the source material (the filmic scenes) is not engaged with analytically. In this sense, it is a very one-sided inquiry into adaptations. This also have consequences for the adaptations themselves. For example, in the adaptation of The Conversation, the author offers very interesting insights into the design-process and the considerations of what to do with the player’s tank in the game. The author decides to remove the player’s tank in order to keep with the two-tank rule of the original Combat: “(…) all the player actually does in this game is listen, so the tank is extraneous (completely obvious when you consider I placed it outside the playing field)", the author notes in their own design documentation (p. 11, figure 7). But if we look at the scene from the film, the protagonist, Harry, is not ‘just listening’ in fact there is a lot going on in this scene: a first shot of Harry standing next to a wall listening to what goes on behind this otherwise opaque wall, a new shot of the hotel room with Harry placing himself in a chair opposite to the wall, the next shot shows the wall from the Harry’s point of view, back at Harry at his chair, back at the wall, but now with Harry’s shadow projected on to it (clearly he has moved back in front of the wall, but is not in the camera’s view), close up of Harry’s face, reverse shot at the wall, shot of Harry walking out on a balcony while tumbling with a light see-through curtain covering the entrance, close-up shot of Harry on the balcony as he is approaching the glass-wall, reverse shot at the actual semi-transparent wall through which the viewer (and Harry!) can see the contours of the murder taking place), close-up of Harry covering his face with his hands, and finally a shot from the inside of the hotel room of Harry entering back into the room and dragging together a thick and completely opaque curtain. As it is clear, this sequence, which lasts less than a minute is extremely tense, and this is reflected not only in Harry’s actions and his anxious body language, but also in the cinematic techniques employed and in the symbolic use of planes of different levels of opaqueness. This is not to say, that the game adaptation should be ‘faithful’ to this particular representation of the event in the hotel room, but more that it would have been nice to see a more reflected analysis of the source film scenes, how they achieve their expressions, and how – and to what extend that can be transferred to the game - rather than just reducing the works to their very basic plot, e.g. a murder is heard, but not seen).
But despite this one sidedness and its lack of engagement with adaptation on a more theoretical and analytical level, I really enjoyed reading the article and the authors rich design reflections. I especially found the reflections on how adaptation might operate even on the level of code, and the constraints of keeping the adaptations within the affordances and representational style of Atari’s Combat truly interesting. The research-through-design approach chosen here, as well as the rich documentation of the design process, may be relevant not only to other game design practitioners but also to adaptation scholars, thus ticking the box of both of the research areas I mentioned in the beginning. The process described in the articles clearly invites – even begs for considerations by adaptation scholars. There already exists intermedial approaches (e.g. Bruhn 2013) that considers adaptations not only in terms of the transfer of plot but also of the different materialities and sensorial experiences afforded by source work and adaption. The current article shows that when considering game adaptations (as well as other code-based media) there is a potential of looking at other material levels, such as the code and hardware.
The article is clearly written with game designers as an audience, but given its potential to inform theoretical and analytical approaches – and given that Arts has a readership that is wider than only game scholars and game designers, I recommend that the author includes in their discussion considerations about possible implications of this experimental design project on the wider area of exchanges between the media.
Finally, the article is well written in nicely illustrated with screenshots from the games. All in all, and based on the above considerations I believe this article, as a reflection upon an original practice based experimental game design project fits nicely with the special issue of Arts and I and recommend that the article should be published.
Author Response
This review was exceedingly helpful and I felt represented a deep engagement with and sympathy toward the objectives in the article. I have changed the article substantially in response - I hope to the reviewer's approval!
As I see it, the article’s most significant contribution is in the area of research-through design, whereas its engagements with of the notion of adaptation remains somewhat superficial.
I address the more specific version of this observation later, but just want to flag here that I acknowledge this particular shortcoming in the paper. While I see it as a situation of not having the resources to be all things to all audience, it is an obvious flaw and is duly noted.
As I read the first part of the this argument, the author sketches out a problem in the state of the art in film-to-game adaptations, which I believe is a result of the economic structures that both the AAA-game and the big-budget film industry is part of. But the author’s solution to this problem is much more formalist in nature exploring and expanding on the expressive potential of games. This then raises the question if such a formal inquiry really poses a solution to a problem that I believe is caused by more economic factors.
I completely agree with this economic/market assessment of the current state of blockbuster-oriented videogame film adaptation (and say as much in the paragraph beginning line 33).
However, I detect a more concerning issue here which is perhaps a small misinterpretation of the key aim of the research project itself. In particular, I do not regard my work in this project as being aimed at solving the problem (to the extent it is one) of Hollywood-leaning videogames, but rather to take inspiration from the idea of videogame film adaptation, currently chronically under-examined in practice, to provide a lens on game design much more generally. Another reviewer also found this unclear, and so it is clear I have been at fault here. I have added to the introduction (line 53) and more extensively to the methodology's introduction (from line 87) to attempt to clarify my stance on this.
First, the chosen films should be adapted not just to games, but to a specific game, namely the Atari classic Combat The reasons for this is that Combat exemplifies games in a ‘pure’ form (in my opinion a somewhat problematic statement), but also more pragmatically, that the game is relatively simple and straightforward in terms of it’s rules and representations.
I completely agree with the flagging of the term "pure" here. I've removed it (line 115), as it did not really serve a useful purpose.
the author, with a few exceptions, does not engage with any existing literature on adaptations (such as Hutcheon 2006; Bruhn et al. 2013; Leitch 2008, 2017,) or specifically about game adaptations (such as Picard 2007, Randall 2016, Flanagan 2017, Svelch 2020 – nor to the many the many works on transmediality such as Tosca and Klastrup 2004, 2014, 2016, Thon and Ryan 2014). This is a shame, because I believe this would also inform the analysis of the author’s own games and their design process. But it is perhaps fair enough, since the author’s focus is clearly on game design (using adaptation mostly as an inspiration or starting point), and not on theorizing (game) adaptations as such, which is clear from both title and contents of the article. Still, I suggest the author to at least clearly delimit the current practice-based inquiry into game design with these more theoretical or analytical contributions.
I am grateful here for the helpful suggestion to be more explicit in the article about my relationship to more theoretical/scholarly approaches to adaptation in general as well as film-to-videogame adaptation in particular. That the reviewer has also generously suggested relevant texts based on their expertise is even more helpful. Starting on line 57 I have included a fairly simple statement with regard to this, including brief citation of work that, while not actively considered in this project, has clear and potentially exciting relevance to my work moving forward. As such, although I think this article in particular is not the place to also reach into the world of cinema studies, adaptation studies, transmedia studies, and more, these are clearly directions to take for future work and I am excited to walk that path.
This is not to say, that the game adaptation should be ‘faithful’ to this particular representation of the event in the hotel room, but more that it would have been nice to see a more reflected analysis of the source film scenes, how they achieve their expressions, and how – and to what extend that can be transferred to the game - rather than just reducing the works to their very basic plot, e.g. a murder is heard, but not seen).
I would just like to note here how wonderful the reviewer's discussion of The Conversation is. I deeply enjoyed the attention to detail all the way down to a shot-by-shot breakdown of the scene. It very clearly shows the chasm between the videogame representation and its source material.
The disjunction between game and cinematic source is of course of interest to me in terms of how it speaks to potential limitations or at least redirections of representation in a game adaptation, but it's equally true that there is a potential missed opportunity involved here. While my process in this project was not to spend significant time in deep analysis of the source material (a task for which I do not feel entirely qualified), I completely agree that this would be a highly valuable approach to take. Indeed, a much larger project can be imagined in which videogame adaptation is much more fully taken to task both in theory and in practice, and this would necessarily include such attention to cinematic language, analysis of specific scenes, the theory of adaptation, and more. (In fact I have a pending grant application to begin just such a pursuit. Here's hoping.)
The article is clearly written with game designers as an audience, but given its potential to inform theoretical and analytical approaches – and given that Arts has a readership that is wider than only game scholars and game designers, I recommend that the author includes in their discussion considerations about possible implications of this experimental design project on the wider area of exchanges between the media.
I very much appreciate this note and have tried to at least indicate in the conclusion (starting at line 544) a hope that there is material of potential interest outside the specific community of game design. Some of my reticence here is related to simply not wanting to make oversized claims for a highly specific work, but I have tried to at least sketch some ideas.